# PEGDA microencapsulated allogeneic islets reverse canine diabetes without immunosuppression

Stephen Harrington[1], Francis Karanu[1], Karthik Ramachandran[1], S. Janette Williams[1], Lisa Stehno-Bittel[1,2]*

1 Likarda, LLC, Kansas City, Missouri, United States of America, 2 Rehabilitation Science, University of Kansas Medical Center, Kansas City, Kansas, United States of America

* lbittel@kumc.edu

## Abstract

### Background

Protection of islets without systemic immunosuppression has been a long-sought goal in the islet transplant field. We conducted a pilot biocompatibility/safety study in healthy dogs followed by a dose-finding efficacy study in diabetic dogs using polyethylene glycol diacrylate (PEGDA) microencapsulated allogeneic canine islets.

### Methods

Prior to the transplants, characterization of the canine islets included the calculations determining the average cell number/islet equivalent. Following measurements of purity, insulin secretion, and insulin, DNA and ATP content, the islets were encapsulated and transplanted interperitoneally into dogs via a catheter, which predominantly attached to the omentum. In the healthy dogs, half of the microspheres injected contained canine islets, the other half of the omentum received empty PEGDA microspheres.

### Results

In the biocompatibility study, healthy dogs received increasing doses of cells up to 1.7 M cells/kg body weight, yet no hypoglycemic events were recorded and the dogs presented with no adverse events. At necropsy the microspheres were identified and described as clear with attachment to the omentum. Several of the blood chemistry values that were abnormal prior to the transplants normalized after the transplant. The same observation was made for the diabetic dogs that received higher doses of canine islets. In all diabetic dogs, the insulin required to attempt to control blood glucose was cut by 50–100% after the transplant, down to no required insulin for the course of the 60-day study. The dogs had no adverse events and behavioral monitoring suggested normal activity after recovery from the transplant.

**Data Availability Statement:** All relevant data are within the paper and its Supporting Information files.

**Funding:** The experiments comprising this study were solely funded by Likarda, LLC.

**Competing interests:** SH and FK are employees of Likarda, LLC. SJW is a former employee of Likarda, LLC. LSB and KR are employees and co-owners of Likarda, LLC. SH and KR are co-inventors on the CSS technology used to create the microspheres tested in this manuscript. Likarda, LLC provided support for the study in the form of salaries for SH and FK. There are no patents, products in development or marketed products associated with this research to declare. Beyond employee salaries, the authors received no specific funding for this work. The competing interests do not alter our adherence to PLOS ONE policies on sharing data and materials.

**Abbreviations:** ANOVA, Analysis of Variance; ALP, Alkaline phosphatase; ALT, Alanine aminotransferase; AST, Aspartate aminotransferase; ATP, Adenosine triphosphate; BW, Body weight; CMRL, Connaught Medical Research Laboratories; $CO_2$, Carbon dioxide; CSS, Core Shell Spherification; DMSO, Dimethyl sulfoxide; DNA, Deoxyribonucleic acid; DPBS, Dulbecco's phosphate buffered saline; DTZ, Dithizone; DVM, Doctor of Veterinary Medicine; EBSS, Earl's Balanced Salt Solution; ELISA, Enzyme-linked immunosorbent assay; FBR, Foreign body response; HEPES, (4-(2-hydroxyethyl)-1-piperazineethanesulfonic acid; IACUC, Institutional Animal Care and Use Committee; IEQ, Islet equivalent; IP, Intraperitoneal; MCH, Mean corpuscular hemoglobin; MCHC, Mean corpuscular hemoglobin concentration; NBF, Neutral buffered formalin; PEG, Polyethylene glycol; PEGDA, Polyethylene glycol diacrylate; RNA, Ribonucleic acid; T1D, Type 1 Diabetes; UV, Ultraviolet light.

## Conclusions and implications

The study provides evidence that PEGDA microencapsulated canine islets reversed the signs of diabetes without immunosuppression and led to states of insulin-independence or significantly lowered insulin requirements in the recipients.

## Introduction

The history of the treatment of type 1 diabetes (T1D) has been highlighted by impressive breakthroughs along with frustrating setbacks. Important breakthroughs include the isolation of insulin, recombinant insulin, insulin pumps, continuous glucose monitors and islet transplants. Islet transplants have been a promising treatment for T1D, but a lack of adequate insulin-producing cells from donors and life-long immunosuppression of the recipients have been two of the major setbacks for the field.

For allogeneic islet transplants, protection from immune rejection is currently provided by systemic pharmacological immunosuppression. However, the reality is that, for many people with T1D, the health risk of pharmacological immunosuppression outweighs the potential for an enhanced quality of life with an islet transplant [1]. Localized immune protection by gel encapsulation holds the promise of transplantation of allogeneic cells without systemic immunosuppression [2]. Coating or encapsulating cells in a hydrogel has long been an approach to mitigate destruction of the transplanted cells [3].

Encapsulation techniques can broadly be categorized as microencapsulation in a bead format and macroencapsulation devices. Both have frustrated the field of cell therapy for diabetes due to some shared and unique challenges. Macro-devices often fail due to insufficient immune-isolation, blood thrombosis for devices that require vascular perfusion, and inadequate diffusion to support cell viability and function [4]. Microencapsulation fails predominantly due to a foreign body response (FBR) to the alginate hydrogel used for encapsulation [5].

Being inexpensive and readily manufacturable, alginate has dominated the cell encapsulation field since its introduction in 1980 [3]. The unique, nearly instantaneous gelation mechanism of the seaweed-derived alginate polymers enables simple fabrication of microcapsules [6]. However, due to the foreign nature of the material, it was notoriously prone to fibrotic overgrowth, which ultimately leads to ischemic necrosis of encapsulated cells, premature graft failure [5–10] and eventually unsuccessful clinical trials [11–13]. Some progress with alginate has been made by developing ultra-purification processes, surface treatments, use of co-encapsulated materials, and more stringent control of capsule microstructure [9, 14–16]. Despite these advances, the performance of alginate microcapsules still does not meet widespread clinical needs [11, 17–19].

Polyethylene glycol (PEG) is a common compound utilized in drug delivery, cosmetics and food [20]. Diacrylated PEG (PEGDA) is a biocompatible, slow-hardening hydrogel that can be used as a replacement to alginate. PEGDA has advantages because the precursors are widely available and inexpensive with appropriate diffusion characteristics [21, 22]. In different formulations, the base chemistry (PEG) has been used to protect islets from immune rejection in multiple studies [23–25]. Unfortunately, because of the slow gelation properties of PEGDA [26], it could not be formulated as microspheres except when produced using emulsion techniques, which can be harmful to cells. Manufacturing utilizing Core Shell Spherification provides a new technology that expands the number and types of hydrogel precursors that can be formed into microspheres [21]. In this study, we encapsulated canine islets in PEGDA

microspheres and characterized the cells and the microspheres. We conducted a pilot biocompatibility/safety study in non-diabetic dogs, transplanting empty microspheres and microspheres containing allogeneic canine islets. The purpose was to determine the biocompatibility of the hydrogel microspheres and to determine whether factors secreated from allogeneic cells within the microspheres could increase the likelihood of a foreign body response. In the subsequent pilot efficacy study, chemically-induced diabetic dogs received increasing doses of encapsulated allogeneic canine islets. Dogs were monitored for blood glucose, weight, blood chemistry, urinalysis, hemotological analysis and pathological/histological assessment.

## Methods

### Islet isolation

Canine islets were obtained locally from donors euthanized for other reasons from local veterinary clinics with consent by owners and the attending veterinarian or from completed research protocols from nonrelated studies. A total of 15 healthy mixed-breed male and female donors were utilized for the study. All dogs were negative for heartworm and were up-to-date in vaccinations. The donor weights varied from 24 to 68 pounds with a mix of breeds including mutts, pointer, German wirehair, beagle, terrier and pit bull mix. All donors were scheduled for euthanasia for other healthcare reasons and were screened for endocrine disorders. The collection of tissue from deceased donors euthanized for reasons other than organ procurement was determined to be exempt by the University of Kansas Medical Center IACUC. Euthanasia was performed by a licensed veterinarian overseeing the care of each animal.

The procurement and digestion protocol followed our previously published methods [27]. The pancreata were removed and transported to the lab in cold conditions where they were trimmed of fat, followed by collagenase digestion. Density gradient purification was performed following our described protocol [27]. Islet equivalents (IEQ) volumes were determined with dithizone staining according to standard published protocols [28, 29]. Canine islets were cultured in CMRL 1066 supplemented with 10% fetal bovine serum, 2 mM glutamine, 10 mM nicotinamide and a 1% antibiotic-antimycotic solution at 37˚C and 5% $CO_2$.

### Canine cell number conversion

We have previously shown that IEQ measurements overestimate the number of cells in a transplant [30]. We published procedures to convert rat and human islets into cell numbers [31, 32]. Here, the procedure was repeated with canine islets, which were cultured in CMRL 1066 supplemented with 10% fetal bovine serum, 2 mM glutamine, 10 mM nicotinamide and a 1% antibiotic-antimycotic solution at 37˚C and 5% $CO_2$. Islets from 3 donors for a total of 450 islets were manually placed in single wells of 384 well plates with 20 μL of media/well (one islet/well). The diameter in the X and Y directions and the total perimeter of each individual islet was measured on a Zeiss AXIO inverted microscope with a Jenoptick C3 camera using Capture Pro V2.8.8 software. Each islet was then dispersed into single cells by exposure to 2 mL of trypsin (0.5%) and manual pipetting. The well contents were mixed and incubated at 37˚C for 1 hour. Individual cells per well were counted on a Biotek Cytation 5 Imaging microplate reader. The data were plotted in a scatter plot to determine the equation to convert canine islet diameters to single cell numbers.

### Critical quality assurance

**Islet purity.** Aliquots of islets from each isolation were tested for purity using our previously published methods [27]. Islets were exposed to dithizone (Sigma Aldrich) prepared fresh

by dissolving dithizone in dimethyl sulfoxide (DMSO) at 10 mg/mL. The solution was diluted to a final concentration of 0.2 mg/mL with PBS. Islets were incubated in the dithizone solution at a 10% v/v ratio. The samples were examined via bright field microscopy for the dark red color resulting from exposure to dithizone. Islet purity was estimated to the nearest 5% for each sample. Isolations with purity below 60% were considered unacceptable and were not used for the transplants.

To verify the accuracy of dithizone staining of canine islets for purity measurement another indicator of insulin, FluoZin-3 AM (Thermo Fisher), was tested. Samples of islets were placed in individual wells of a 384 well plate and dispersed into single cells as described above. Cells were exposed to FluoZin-3 AM (1 μL) added to each well and incubated at 37˚C for 1 hour. Images were collected on a Biotek Cytation 5 Imaging Multi-Mode microplate reader and the number of FluoZin-positive cells counted and used to calculate the % insulin positive cells by the following equation:

$$\text{FluoZin–positive cells} / (\text{FluoZin–positive cells} + \text{FluoZin–negative cells})$$

**Viability.** Prior to transplantation, encapsulated islets were stained for viability. Cells were incubated in calcein AM (4 μM, ThermoFisher) and propidium iodide (1 μg/mL, ThermoFisher) for 30 minutes. Fluorescence was captured with a Cytation 5 Imaging Multi-Mode Reader (Biotek Instruments). Using the Cytation software, the area of calcein-stained (live) cells was divided by the total cell area (obtained with brightfield images of the same fields) resulting in the percentage of live cells. Following encapsulation, microspheres were incubated in calcein AM (4 μM, ThermoFisher) and propidium iodide (1 μg/mL, ThermoFisher) for 40 minutes and the same procedures followed as described for unencapsulated cells.

**Insulin secretion.** Glucose stimulated insulin secretion was conducted using our previously published protocol [27, 33]. Briefly, the unencapsulated canine islets were exposed to glucose solutions based on Earl's Balanced Salt Solution (EBSS) buffer with 0.1% BSA and sodium bicarbonate added (pH 7.4, 37˚C at 5% $CO_2$). Islets were first equilibrated to the low glucose condition of 2.8 mM for 1 h. Transwell inserts in a 24-well plate held approximately 22,000 cells and were used to transfer the islets to increasing concentrations of glucose. Supernatant media was collected after 1 hour in each concentration and stored at −80˚C until quantification was performed using Mercodia's canine insulin ELISA kit.

To measure glucose stimulated insulin secretion from encapsulated cells, approximately 10 microspheres were incubated as described above to equilibration in low glucose. Rather than the transwell procedure, different microspheres were incubated in 2.8, 16.7 or 28 mM containing 30 mM $K^+$ as a secretagogue for 90 minutes.

**Insulin content.** Total protein of islet aliquots was extracted by acid ethanol (0.18 MHCl in 95% ethanol) as we have published previously [34]. Briefly, the total insulin content was determined by canine insulin ELISA (Mercodia). Insulin content was normalized to cell number as we have done previously [35].

**DNA and ATP content.** To isolate total DNA from islets, the GeneJET Genomic DNA Purification Kit (ThermoFisher Scientific) was used following the manufacturer's recommendations. Briefly, a predetermined number of islets was incubated in 180 μL of digestion solution and 20 μL proteinase K for 1–3 hrs in a 37˚C shaking water bath followed by RNAse treatment. Total DNA was then isolated by cell lysis and a series of ethanol washes. Eluted DNA was then quantitated using the Quant-iT PicoGreen dsDNA Assay Kit (ThermoFisher Scientific) according to manufacturer's instructions.

ATP levels were measured using a luminescent ATP assay (CellTiter-Glo; Promega) following the manufacturer's instructions. Islets were distributed in 96 well plates in CMRL media.

They were exposed to the CellTiter-Glo reagent, and 15 min later luminescence read on a Cytation 5 Imaging Multi-Mode Reader.

**Encapsulant manufacturing.**   PEGDA microspheres were manufactured following our previously published Core Shell Spherification (CSS) protocol [21]. Briefly, PEGDA hydrogel precursor solution was prepared by dissolving PEGDA 3,400 and 20,000 (Laysan Bio, Inc.) at 18% and 12% (w/w), respectively, in a buffer containing 100 mM calcium chloride, 10 mM HEPES, and 0.025% (w/v) Irgacure 2959 followed by filtering through a 0.22-mm syringe. The viscosity of the precursor was measured with a Cannon-Manning Semi-Micro calibrated glass capillary viscometer at room temperature. The precursor was extruded via an automated drop-let generator; a Buchi 395-Pro Encapsulator (Buchi Corporation, Newcastle, DE) equipped with an air jet nozzle system and a 400-micron diameter inner fluid nozzle within a 1.5 mm concentric air nozzle. The droplets fell into a bath of 0.15% (w/v) sodium alginate (Protanal LF 10/60, FMC Corp), Irgacure 2959 (0.025% w/v), 0.1% Tween 20, mannitol (300 mM), and adjusted to pH 7.6. The PEGDA droplets were irradiated with long-wave UV light of 40 mW/$cm^2$ at the center of the bath (PortaRay 400, Uvitron International) through extrusion and for an additional 1 minute. The resulting microspheres were rinsed twice in a 25 mM citrate buffer in DPBS for 5 minutes and collected using a steel mesh screen suspended in DPBS. Empty PEGDA microspheres were cultured and stored at 24°C and 5% $CO_2$. Microspheres were transported in suspension in cold storage solution in a sealed sterile vessel at 2–8°C until administration.

The islet-containing microspheres were fabricated as described above except that the canine islets were mixed into the precursor at a 10:1 volume ratio just prior to droplet generation. Fabrication of islet-microspheres was done aseptically in a closed, sterile bioreactor system as described above.

**Recipient dogs.**   Seven beagles were obtained from Marshall or Ridglan Farms. The non-diabetic dogs were maintained at Veterinary & Biomedical Research Center and the diabetic dogs at Sinclair Research. Both organizations approved the full protocols and addendums through their IACUCs. All dogs were acclimated at the study site for a minimum of 10 days. Dogs were housed in a minimum of 8x8 runs with raised floors, maintained at 64–84°F and a 12–12 dark/light cycle. Dogs were monitored by an automated room building surveillance sys-tem as well as by video access 24/7. They were fed commercial dog chow, twice daily with free access to water.

**Diabetes induction.**   Diabetes was induced in 3 of the 7 dogs with a single injection of a combination of alloxan (40 mg/kg) and streptozotocin (30 mg/kg) IV. Subsequently blood glu-cose levels were maintained with twice daily insulin (Vetsulin) injections during the study period. Blood glucose values were measured at least twice daily for each animal using a porta-ble glucometer.

**Transplant procedure.**   The microspheres were placed into the omental tissue via a lapa-rotomy under general anesthesia. The first 4 non-diabetic dogs (safety study) received half of the dose with empty microspheres on one side of the omentum and the other half were islet-containing microspheres were infused on the other side. In this manner, the effect of secretory factors from the cells could be separated from a possible foreign body response to the hydrogel. In addition, we examined the blood glucose for any negative effect of the allogeneic islets or hydrogel on blood chemistry and hematology. For the second efficacy portion of the study, 3 diabetic dogs received escalating doses of islet-containing microspheres with monitoring as described above.

Animals were anesthetized either endotracheally or with a mask using isofluorane (0.5% to 5% in 100% oxygen). The target area was prepped using standard surgical procedures. The midline abdominal incision was made, and the subcutaneous tissue dissected longitudinally

and the linea alba identified. The peritoneum was exposed with a large enough opening to fit the catheter for direct infusion. The tip of the catheter was placed near or touching the omentum and the encapsulated cells infused into the area frequently repositioning the catheter to cover the surface with microbeads. The animals were monitored closely during surgical procedures and observed until fully recovered from anesthesia post-surgery.

**Post-transplant monitoring.** General in-cage observations for mortality/moribundity were made at least twice daily. Physical examinations included the skin (particularly abdomen) and external ears, eyes, abdomen, behavior, and general body condition. Dogs were weighed weekly. Throughout the monitoring period, blood chemistry, urinalysis, and hematological analysis were conducted weekly or every 2 weeks by an independent reference laboratory. Blood glucose was monitored 2/day initially after the transplant reducing to daily or every other day as dog blood glucose normalized. Hypoglycemia was defined a < 60 mgdL [36]. Hyperglycemia was defined as > 250 mg/dL [37] with diabetes defined as persistent non-fasting hyperglycemia for 1 week.

**Study termination.** Animals were humanely euthanized with Fatal Plus, following sedation with Telazol and Xylazine. A complete necropsy was performed on each animal by a supervising pathologist, which included examination of the external surfaces, all orifices, the cranial, thoracic, abdominal and pelvic cavities, including contents. The implantation site and internal organs were retained for subsequent tissue preservation and histopathological analysis.

Samples of tissues surrounding the transplant site as well as other organs were preserved in 10% neutral buffered formalin (NBF), except for the eyes and optic nerve, which were placed in Davidson's solution for 1–3 days and then stored in 10% NBF. Tissues were embedded in paraffin, sectioned and stained with hematoxylin and eosin. The slides were evaluated by a DVM pathologist.

## Statistics

One-way ANOVA with Tukey's test for post-hoc analysis was used for all statistical comparisons with $p < 0.05$.

## Results

### Canine islet characterization

**Islet dosing.** Cell therapies are predominantly dosed based on the cell number, but in the islet transplant field, IEQs are still used. There are significant differences in the average number of cells per IEQ between species [31, 32]. Thus, it was important to use our previously published protocol [31, 32] to determine the average number of cells in canine islets of different diameters. Fig 1A illustrates the correlation between the diameter of canine islets and the number of single cells within each islet (R = 0.75). The data used to create the graph are provided in S1 Table in S1 File. Based on the values, a canine islet of 150 μm diameter (one IEQ) would contain an average of 430 cells, which is dramatically lower than the number of cells/IEQ for human or rat islets [31, 32]. The correlation between size and cell number was used to calculate dosing of the subsequent transplants. The majority of isolations were predominantly comprised of islets under 50 μm in diameter (Table 1; Data provided in S2 Table in S1 File).

**Islet purity.** For batches of intact islets that were used in the transplants, dithizone staining was used to determine the purity of the sample (Fig 1B). The dithizone-based purity of the dog islets ranged from 41–65% and is provided for each transplant in Table 1. To verify the results, additional samples were exposed to FluoZin 3 AM (Fig 1C). On average 40–45% of the cells within each canine islet sample were FluoZin 3 AM positive (β-cells) (Table 2 and S3

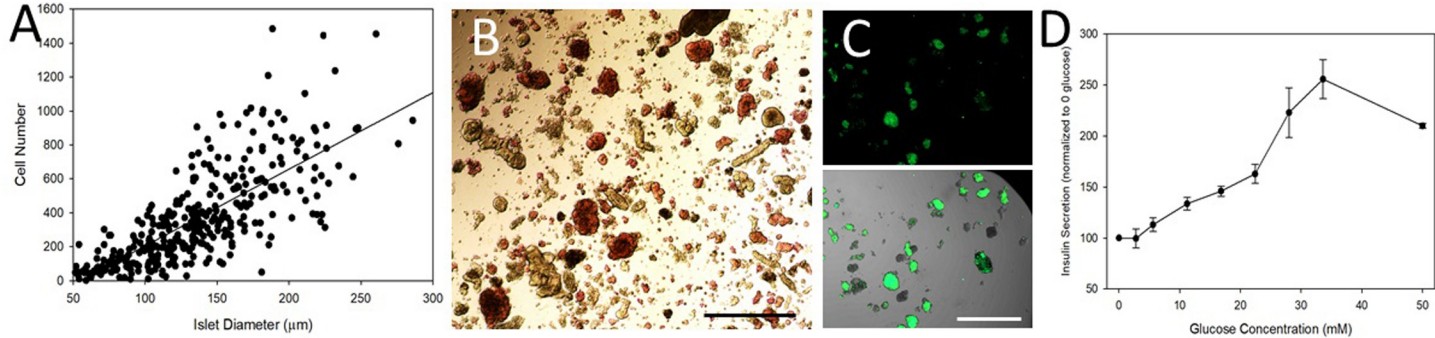

**Fig 1. Characterization of canine islets from deceased donors.** A) Diameters of 450 freshly-isolated canine islets were measured in the X and Y direction, and subsequently dispersed into single cells, which were counted. The correlation between the canine islet diameters and cell numbers were plotted. The line indicates the linear fit. B) An example of dithizone-stained canine islets with red staining the endocrine cells and the non-cells (tan) indicating exocrine tissue. Scale bar = 500 μm. C) FluoZin 3 AM was used to further identify canine islets, shown as bright green in the images. The upper image is fluorescence only while the lower image is a brightfield overlay. Scale bar = 500 μm for all images. D) Static incubations studies of canine islets showed that they responded to increasing concentrations of glucose with higher levels of insulin secretion.

Table in S1 File). The lower image shows an overlay of both the fluorescence and the bright field images used to calculate the total cellular area.

**Islet viability.**   Live-dead staining was conducted on islets prior to encapsulation (S4 Table in S1 File) and after microencapsulation for the last 5 batches and ex vivo when sufficient loose microspheres could be obtained from the last dog (S5 Table in S1 File). Cells were stained with calcein (green) identifying live cells and PI (red) identifying dead cells. Viability of the cells prior to encapsulation ranged from 71 to 89% (Tables 1 and 2). There was no statistical difference in the pre- or post-encapsulation viability of the islets infused into healthy or diabetic dogs (Tables 1 and 2).

**Functional characterization.**   Canine islets were assayed for stimulated insulin secretion using static incubation techniques. Islets were exposed to the following glucose concentrations: 0, 2.8, 5.6, 11.2, 16.8, 22.4, 28, 33.6 and 50 mM glucose for one hour (S6 Table in S1 File). When exposed to increasing concentrations of glucose, the islets secreted more insulin up to 50 mM glucose, which induced a slight decline in secreted insulin compared to 33.6 mM (Fig 1D), similar to values obtained previously for canine islets [27].

The islets transplanted into the three diabetic dogs (recipients 5–7) underwent additional characterization. Aliquots were used for ATP and insulin determinations with the results shown in Table 2. The values were calculated as the amount per cell using the cell number conversion shown in Fig 1A and S7 Table in S1 File. Cellular ATP and insulin levels were not statistically different across transplant batches.

**Encapsulated islets.**   Canine islets were encapsulated in a PEGDA formulation resulting in microspheres averaging 953.4 + 8.2 μm in diameter (Fig 2A). The average microsphere diameter for each batch is found in Tables 1 and 2 with the raw data provided in S8 Table in S1

**Table 1. Islet characteristics for biocompatibility/safety study.**

| Recipient ID | Purity (% β-cells) | Size Distribution (% < 50 μm diameter) | Viability pre-encapsulation (% live) | microsphere size (μm) | Average islet clusters per microsphere |
|---|---|---|---|---|---|
| 1 | 45 + 4 | 77.3 | 75.4 | 1054 | 12.1 + 0.5 |
| 2 | 60 + 2 | 43.0 | 72.1 | 1033 | 12.3 + 0.8 |
| 3 | 49 + 5 | 74.7 | 71.5 | 930 | 12.6 + .06 |
| 4 | 52 + 6 | 74.7 | 71.0 | 930 | 13.2 + 0.7 |

**Table 2. Islet characteristics for efficacy study.**

| Dog ID | Purity (% β-cells) | Size (% < 50 μm diameter) | Viability pre-encapsulation (% live) | Viability post-encapsulation (% live) | ATP pM/cell | Insulin pg/cell | Islet Clusters/ sphere | Sphere diameter (μm) |
|---|---|---|---|---|---|---|---|---|
| 5 | 41 + 2 | 69.8 | 70.6 | 69.2 | 0.021 | 11.3 | 15.8 + 1.6 | 1092 |
|  | 65 + 5 | 75.8 | 76.0 | 69.3 |  |  | 18.8 + 0.8* | 960 |
| 6 | 49 + 2 | 74.4 | 88.9 | 66.5 | 0.018 | 15.9 | 14.7 + 0.6 | 911 |
| 7 | 49 + 3 | 84.4 | 76.5 | 74.9 | 0.029 | 10.9 | 16.5 + 1.2 | 991 |
|  | 65 + 5 | 75.8 | 76.0 | 69.3 |  |  | 15.2 + 0.7 | 960 |

*$P < 0.001$.

File. Dithizone staining of the encapsulated cells identified the insulin-positive islets contained within the microspheres (Fig 2B) but could not be used to determine purity values after encapsulation, due to the 3D nature and the blunted dithizone staining after encapsulation. Counts were made of the number of dithizone-stained clusters per microsphere (S9 Table in S1 File). There were no statistical differences in the number of islet clusters/ sphere administered to the healthy dogs (Table 1). However, in the transplants given to the diabetic dogs, the islets/sphere was higher for the second transplant for dog #5 compared to all other groups (Table 2).

Viability measurements after encapsulation were not collected for the first four dogs in the biocompatibility study. Fig 2C provides an example of the viability staining following microencapsulation. Post-encapsulation viability remained high between 67 and 75% for the transplants into diabetic dogs (Table 2 and S5 Table in S1 File). With encapsulation, the death of single cells was noted, while intact islets continued to thrive in the PEGDA formulation.

Insulin secretion of the encapsulated cells was measured with static incubation (S10 Table in S1 File). Fig 2D summarizes the increase in insulin secretion with exposure to high glucose or high glucose with a secretagogue (high $K^+$).

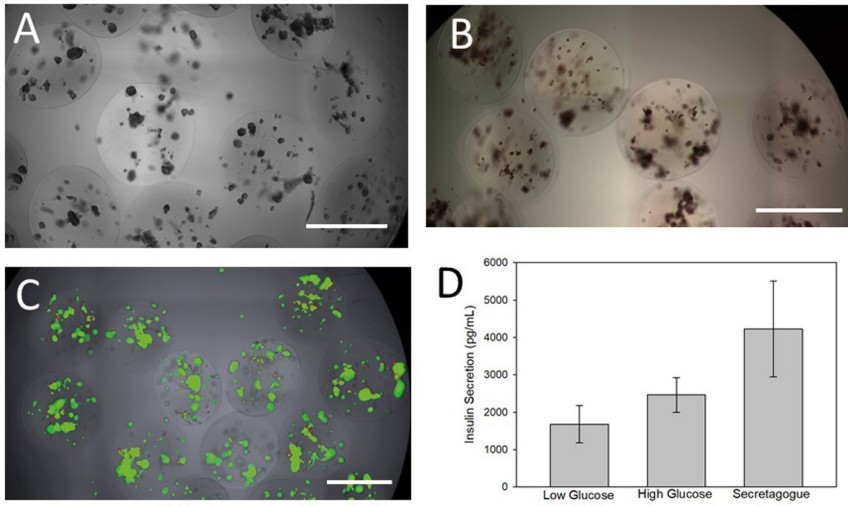

**Fig 2. Characterization of microencapsulated canine islets.** A) Typical field of microspheres containing canine islets. B) Dithizone staining of the encapsulated cells indicated the insulin-positive islets contained within the microspheres. C) Viability was determined with calcein and PI showing a typical example. Scale bars = 1 mm for all images. D) Static incubation insulin release was measured from encapsulated islets in low (2.8 mM), high (16.7 mM) and high glucose with 30mM $K^+$ (secretagogue).

### In life data

**Canine transplants.** To test the biocompatibility of the PEGDA microspheres in dogs, four healthy dogs received encapsulated islets on one side of the omentum with empty microspheres injected into the opposite side. Using this method, if a foreign body response was noted, the effect of the hydrogel could be separated from the effect of the cells plus hydrogel. In a subsequent efficacy study, three diabetic dogs received increasing doses of encapsulated islets into the omentum. Table 3 provides a summary of the recipients including the total islet dose (cells/kg body weight). The procedure, using a catheter to infuse the encapsulated cells is shown in Fig 3A. After administration, a portion of the omentum was exposed in one animal showing the microspheres sticking to the tissue (Fig 3B).

All of the dogs tolerated the procedure well based on daily observations of attitude, appetite scores, urination scores, water consumption, and rectal temperature, which were all normal for the duration of the two studies with an average body temperature of 100.06. In the non-diabetic dogs, there was little change in the average weekly body weight after their transplants (Fig 4A). For the diabetic dogs, there was a slight increase in weekly body weight after the transplants (S11 Table in S1 File). This is important as diabetic dogs typically lose weight when treated with insulin only.

**Blood glucose.** For the first four healthy dogs that received microspheres with half of the spheres containing islets, there were no measured hypoglycemic events. Weekly fasting blood glucose values are shown in Fig 4B with the supporting data in S12 Table in S1 File. This was even true for dog #4 that received the highest dose of cells for the healthy dogs (1.7 M cells/kg) (Table 3).

For the diabetic group, dog #5, received an escalating dose of encapsulated islets in two separate administrations for a total of 5.1 M cells/kg. The first dose was approximately 1.46 M cells/kg and normalized blood glucose with only half of the dose of insulin required compared to the pre-transplant levels (Table 4). The second dose was 3.69 M cells/kg, which maintained blood glucose in the target range for more than 3 weeks (Fig 4C with supporting data in S13 Table in S1 File). However, after that time, the blood glucose values steadily increased and daily insulin injections were resumed, although requiring approximately 1/6 the insulin dose required prior to the transplant (Table 4). Fasting blood glucose post transplantation show the gradual increase in blood glucose to the termination of the study (Fig 4B).

Recipient #6 received a single administration at a higher total of 6.2 M cells/kg (Table 3). There was an initial decline in blood glucose into the target range, but days later the average daily non-fasted blood glucose was over 300 mg/dL (Fig 4C). Prior to the transplant, the dog required an average of 11 units of insulin/day and was still hyperglycemic most of the time with an average blood glucose of 330 mg/dL. After the transplant the insulin requirement was reduced to 6 units of insulin daily (Table 4). Fasting blood glucose readings over the course of the study illustrated the lack of long-term effect with the single dose (Fig 4B).

**Table 3. Transplant description.**

| Dog ID | Disease Status | Weight | Sex | Age (years) | Dose (cells/kg bw) | Duration post-transplant |
|--------|----------------|--------|-----|-------------|--------------------|--------------------------|
| 1 | Healthy | 11.0 | F | 4 | 205,090 | 28 |
| 2 | Healthy | 10.0 | F | 4 | 225,670 | 28 |
| 3 | Healthy | 13.6 | F | 4 | 595,000 | 14 |
| 4 | Healthy | 12.7 | F | 4 | 1,679,500 | 14 |
| 5 | DM | 9.4 | M | 7 | 5,145,083 | 63 |
| 6 | DM | 9.5 | M | 7 | 6,238,850 | 65 |
| 7 | DM | 7.4 | M | 7 | 7,370,575 | 58 |

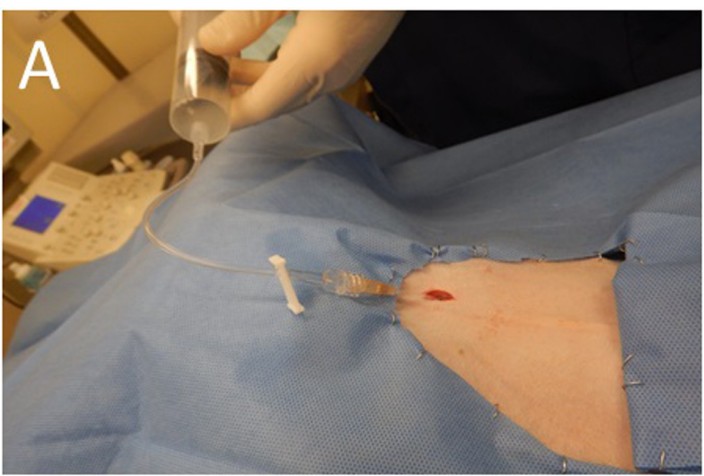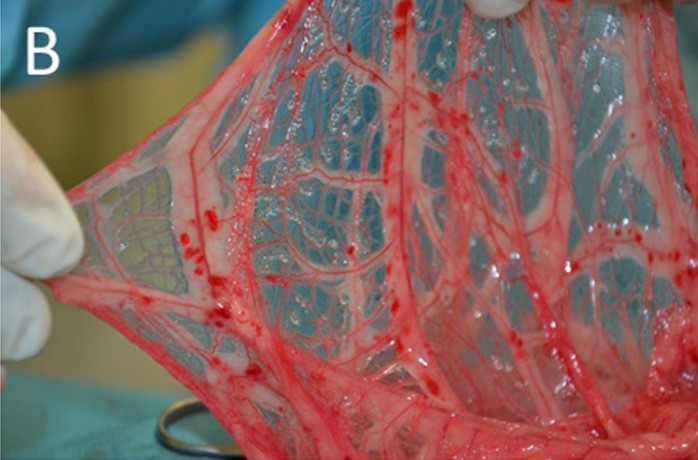

**Fig 3. Infusion procedure.** A) Encapsulated islets were infused into the peritoneum of the dogs using a sterile syringe and catheter. B) After infusion, a portion of the omentum was retrieved illustrating the adherence of the microspheres to the omentum, which was immediately returned to the peritoneum.

Dog #7 was dosed at the highest level with a total transplant of 7.4 M cells/kg administered in two procedures. After the first transplant, the blood glucose fell to the target range for most of the following 3 weeks (Fig 4C). Prior to the first transplant, the dog required an average of nearly 13 units of insulin/day (Table 4). After the first transplant, the insulin requirement averaged less than 1 unit per day. Following the second transplant no further insulin was administered and the blood glucose normalized and maintained those levels through termination of the study. The fasting blood glucose values illustrate the improved blood glucose regulation following the second transplant (Fig 4B with supporting data provided in S12 Table in S1 File).

While the dose of the transplants (cells/kg recipient body weight) tracked with the final blood glucose levels, we calculated other cellular characteristics to determine if they also tracked with the final week's average blood glucose levels (Table 5). The cellular DNA per kg body weight of each recipient can serve as a surrogate for cell number. Acknowledging that three animals is not sufficient to draw statistical conclusions, none of the calculated cellular measurements (DNA, ATP, or insulin levels) tracked as well as the cells/body weight in this pilot study.

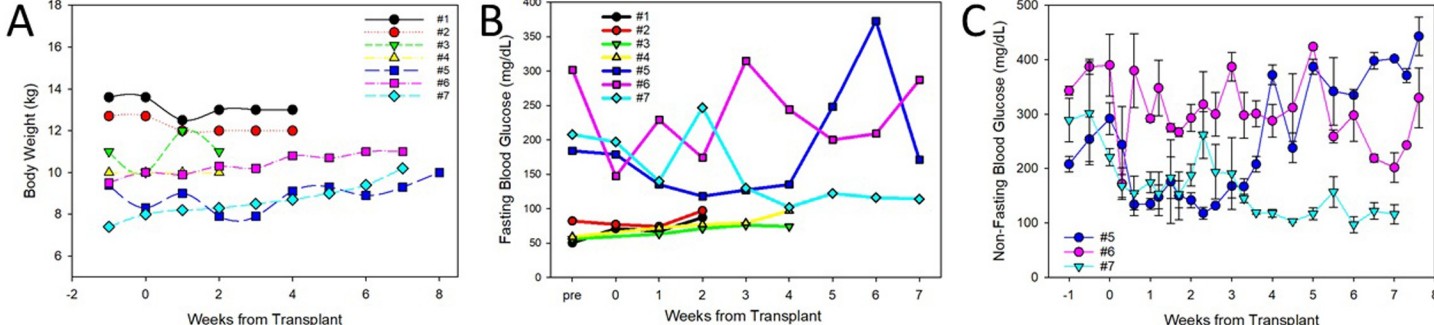

**Fig 4. Body weight and blood glucose values.** A) Body weight was obtained weekly showing little change over the course of the two studies. B) Fasting blood glucose was monitored weekly in the animals. Dogs #1–4 showed no change in blood glucose. Dog #5 had the lowest total dose of islets among the diabetic dogs that were administered in 2 doses; the first on day 0 and the second 2 weeks later. He had a temporary return to blood glucose levels within the normal range. Dog #6 had a higher dose, but administered at one time. This dog had little response to the transplant. Dog #7 had the highest dose again administered in 2 rounds and showed a full response with no further requirement for exogenous insulin. C) More detail in the daily fluctuations of the non-fasting blood glucose levels is provided for the 3 diabetic dogs.

**Table 4. Insulin utilization.**

| Dog ID | Disease Status | Number of Transplants | Timing of Transplants (Day) | Total Cells Transplanted | Average insulin pre-transplant | Average insulin post-transplant |
|---|---|---|---|---|---|---|
| 1 | Healthy | 1 | 0 | 2,235,481 | 0 | 0 |
| 2 | Healthy | 1 | 0 | 2,662,906 | 0 | 0 |
| 3 | Healthy | 1 | 0 | 7,021,000 | 0 | 0 |
| 4 | Healthy | 1 | 0 | 21,329,650 | 0 | 0 |
| 5 | DM | 2 | 0 | 13,701,318 | 5.8 + 0.7 | 2.0 + 0.5 |
| | | | 14 | 34,662,469 | | 1.1+ 0.2 |
| 6 | DM | 1 | 0 | 59,269,051 | 11.0 + 1.3 | 6.1 + 0.2 |
| 7 | DM | 2 | 0 | 19,879,789 | 12.9 + 0.8 | 0.6 + 0.1 |
| | | | 21 | 34,662,469 | | 0.0 + 0.0 |

**Blood chemistry.** Blood chemistry analysis was conducted by a third-party reference laboratory. Only the abnormal values are included in Table 6. The full blood chemistry results are presented in S14 Table in S1 File. The tables include the values prior to the first transplant, 1 week following the first transplant and at the termination of the study. It is important to recognize that the duration was different for each group (Table 3). For several factors, the transplants normalized blood chemistry values, if not bringing them into the normal range. For example, dogs #1–4 had abnormally high creatinine kinase prior to the implantation procedure, but by the study end, the values had all normalized. Similar trends of normalization of values that were out-of-range prior to the transplants were identified for creatine kinase, globulin, total protein, calcium, and potassium. The liver enzymes ALP, AST and ALT were high in all of the diabetic dogs (#5–7) but normalized after the transplants. However, dog #1 responded in the opposite manner with normal or low values prior to the transplant and elevated values after. Although he had no overt clinical signs of liver dysfunction through the study. Dog #1 had received the lowest dose of islets, thus, it seems unlikely that the reaction was due solely to the transplant.

**Hematology and urinalysis.** Hematology analysis was normal for all readings with the exception of those shown in Table 7 (complete hematology values included in S15 Table in S1 File). Slightly low Mean corpuscular hemoglobin (MCH) was noted in the diabetic dog #7 prior to the transplant, which normalized by the termination of the study. When the values were calculated as concentrations all the dogs showed low levels of mean corpuscular hemoglobin concentration (MCHC), but again normalized by the end of the study. The same was true for the eosinophil count for dog #4. Only dog #5 had a normal value that became abnormal after the transplant. His monocytes count was slightly elevated 1 week after the transplant, but again normalized by the end of the study. All other hematological values were within the normal range. Urinalysis revealed some blood in the urine in 2 dogs before the transplants, which did not change after transplantation. Analysis of coagulation factors revealed the fibrinogen levels rose in 5 dogs after the transplants, but by study ends, only 2 dogs had values above the normal range.

**Table 5. Transplant cellular characteristics.**

| Recipient ID | Cell number/kg | DNA μg/kg | ATP nmol/kg | Insulin μg/kg | Average BG (final week) |
|---|---|---|---|---|---|
| 5 | 5.1 | 242.7 | 124.0 | 75.6 | 405 |
| 6 | 6.2 | 369.5 | 110.5 | 99.3 | 248 |
| 7 | 7.4 | 356.5 | 240.3 | 97.6 | 114 |

**Table 6. Abnormal blood chemistry values.**

| Test | Normal Range | Dog ID | Pre | 1 wk post | Termination |
|---|---|---|---|---|---|
| ALP (U/L) | 5–160 | 1 | 10 | 65 | *276* |
| | | 5 | *162* | 94 | 101 |
| | | 6 | *175* | 108 | 83 |
| | | 7 | *224* | 123 | 105 |
| AST (U/L) | 16–55 | 1 | 49 | *68* | *184* |
| ALT (U/L) | 18–121 | 1 | *8* | *268* | *731* |
| | | 5 | *184* | 87 | 79 |
| | | 6 | *148* | 45 | 30 |
| | | 7 | *439* | 98 | 63 |
| Creatine kinase (U/L) | 10–200 | 1 | *534* | *274* | 150 |
| | | 2 | *586* | *238* | 135 |
| | | 3 | *285* | 194 | 137 |
| | | 4 | *746* | *346* | 141 |
| Total bilirubin (mg/dL) | 0.0–0.3 | 1 | *0.4* | *0.5* | *0.5* |
| | | 2 | *0.6* | 0.3 | 0.2 |
| | | 4 | *0.7* | *0.5* | 0.3 |
| Bilirubin conjugated (mg/dL) | 0.0–0.1 | 1 | 0.0 | *0.2* | 0.1 |
| Bilirubin unconjugated (mg/dL) | 0.0–0.2 | 1 | *0.4* | 0.2 | *0.6* |
| | | 2 | *0.5* | 0.2 | 0.1 |
| | | 4 | *0.6* | 0.2 | 0.1 |
| Globulin (g/dL) | 2.4–4.0 | 2 | *2.2* | 3.2 | 3.0 |
| Total Protein (g/dL) | 5.5–7.5 | 2 | *5.2* | 6.0 | 6.0 |
| Creatinine (mg/dL) | 0.5–1.5 | 2 | 0.5 | *0.4* | *0.4* |
| | | 4 | *0.3* | *0.4* | *0.4* |
| | | 6 | *0.4* | 0.7 | 0.8 |
| Calcium (mg/dL) | 8.8–11.2 | 1 | *7.5* | 9.0 | 9.7 |
| | | 2 | *6.4* | 8.9 | 10.0 |
| | | 3 | *7.3* | 9.6 | 10.3 |
| | | 4 | *6.2* | 9.3 | 10.1 |
| Phosphorus (mg/dL) | 2.5–6.1 | 7 | 6.5 | *6.7* | 5.4 |
| Chloride (mmol/L) | 108–119 | 3 | *107* | 108 | *107* |
| | | 4 | *107* | *107* | *104* |
| | | 5 | 109 | *106* | *105* |
| | | 6 | 108 | *107* | *104* |
| Potassium (mmol/L) | 4.0–4.5 | 1 | *5.5* | *5.1* | *5.1* |
| | | 2 | *5.0* | *4.6* | 4.4 |
| | | 3 | *5.0* | *4.7* | 4.5 |
| | | 4 | *5.3* | *4.7* | 4.4 |
| | | 5 | *4.7* | 4.0 | *5.1* |
| | | 6 | *3.9* | 4.1 | 4.4 |

**Pathology results.** At the termination of the study, the administration site (the omentum) was exposed and attached microspheres were identified (Fig 5A and 5B). The histopathological analysis of organs and tissue from dogs #3 and #4 found that 14 days after the transplants the dogs had moderate multilocular inflammatory responses. One dog showed signs of a foreign body response surrounding some of the microspheres on day 14 (Fig 6A). By day 28, the inflammatory response was decreased and described as mild in dogs #1 and #2 (Fig 6B). The

**Table 7. Hematology.**

| Test | Normal Range | Dog ID | Pre | 1 week post | Termination |
|------|--------------|--------|-----|-------------|-------------|
| MCH | 21.9–26.1 | 7 | *21.8* | *21.8* | 23.3 |
| MCHC | 32.6–39.2 | 5 | *31.2* | *31.3* | 38.4 |
| | | 6 | *32.3* | *31.9* | 33.6 |
| | | 7 | *31.5* | *31.4* | 33.6 |
| Monocytes | 130–1150 | 5 | 750 | *1200* | 940 |
| Eosinophils | 70–1490 | 4 | *54* | 123 | 190 |
| Coagulation–Fibrinogen | 90–255 | 2 | 189 | *393* | 241 |
| | | 3 | 151 | *352* | 214 |
| | | 5 | *335* | *377* | *324* |
| | | 6 | 196 | *382* | 186 |
| | | 7 | 195 | *276* | *261* |

inflammatory cells noted surrounding the microspheres at 14 days (Fig 6A) were absent at 28 days and only a minimal ring of fibrotic tissue surrounded the microspheres within healthy omentum (Fig 6B). Importantly, there were no differences noted between the portions of the omentum that had cell-containing microspheres versus areas with empty microspheres (Fig 6C and 6D).

In the diabetic animals, the islet-containing microspheres were identified within the omentum and attached to the surface of the liver and the body of the pancreas. The other organs examined had no identified microspheres and were classified as unremarkable. Those included the mesenteric lymph nodes, the large and small intestines, head and tail of the pancreas, spleen, testes, and bladder. Mild fibrosis was identified in dog #5 within the omentum surrounding the microspheres (Fig 7A). Dog #6 had only minimal fibrosis around some of the microspheres (Fig 7B). Dog #7 had a few microspheres attached to the liver that were surrounded by granulomatous inflammation and fibrosis (Fig 7C). All other sites examined were normal. An important observation was the severe foreign body response that was noted when microspheres were clustered together, rather than separated. While the vast majority of the

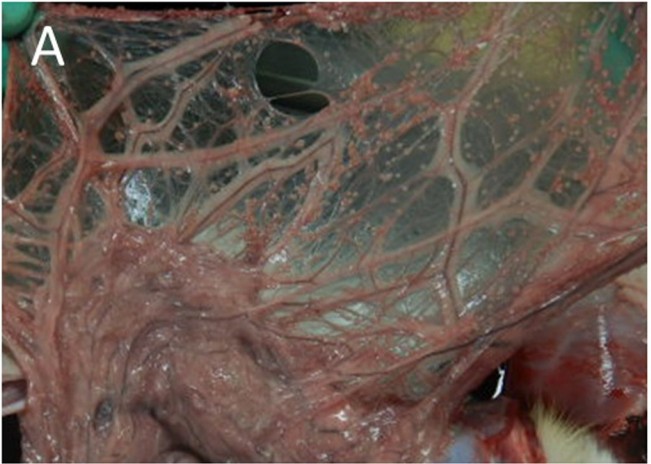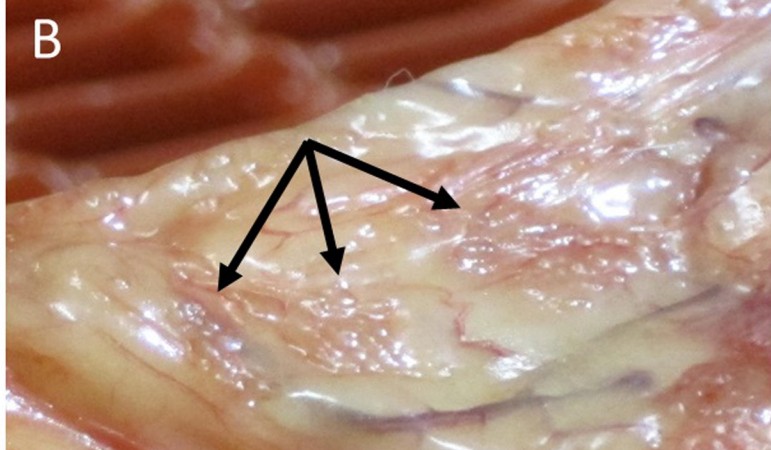

**Fig 5. Necropsy images of microspheres attached to the omentum.** A) The retrieved omentum is shown from dog #7 with microspheres scattered throughout the tissue. B) A higher magnification of the tissue from another animal illustrates the location of the microspheres (black arrows) and the surrounding blood supply with healthy omentum.

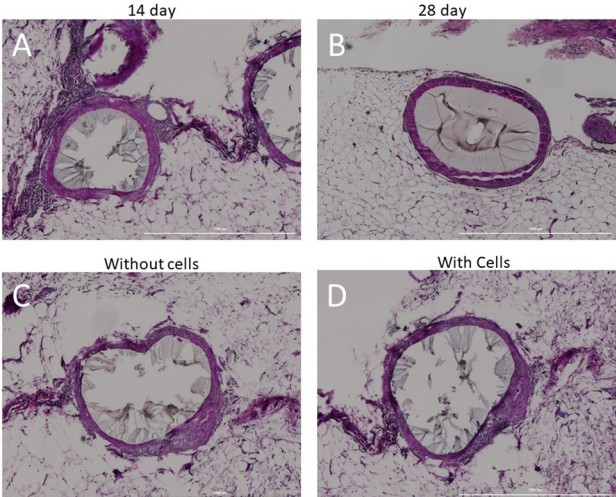

**Fig 6. Histology of transplant sites in healthy dogs.** A) Example of the mild peri-microspheres fibrosis with inflammatory cells in the surrounding tissue was noted in healthy dogs at 14 days. B) By 28 days the inflammatory response had decreased and was pathologically described as mild. C) Example of foreign body response surrounding microsphere without cells. D) Example of response to microsphere containing allogeneic canine islets. All scale bars = 1000 μm.

transplanted material consisted of individual spheres, when microspheres were found near each other, the foreign body response was exaggerated (Fig 7D).

**Explanted microspheres.** The majority of microspheres were attached to the omentum, however, some free microspheres were retrieved and were found to be void of any cellular overgrowth or fibrosis as can be noted in Fig 8A. Encapsulated cells were stained for viability using calcein (green) for live cells and propidium iodide (red) for dead cells (Fig 8A) demonstrating live islets retrieved from the omentum. A higher magnification image from another

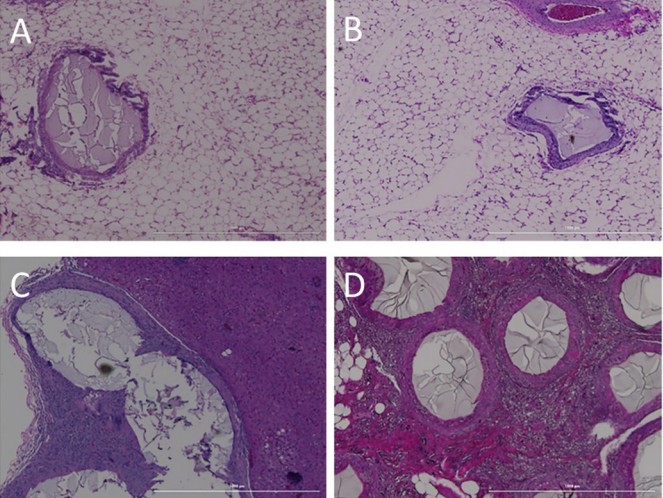

**Fig 7. Histology of transplant sites int diabetic dogs.** A & B) In diabetic dogs with longer time points (58–65 days) any indication of a foreign body response in the omentum was classified as mild with incomplete fibrotic rings. C) However when a few microspheres attached to the liver, granulomatous inflammation and fibrosis were noted. D) Microspheres within the omentum, but clustered together, also elicited a stronger foreign body response than individual microspheres. All scale bars = 1000 μm.

microsphere revealed a few dead cells within the individual islets (Fig 8B). The average viability of the canine islets prior to encapsulation, after encapsulation and after transplantation (explants) were compared in Fig 8C with supporting data provided in S5 Table in S1 File. There were no statistical differences between the groups.

Dithizone staining of the encapsulated islets were used to compare the number of insulin-positive islet clusters prior to transplant and in the explanted microspheres at the termination of the study (Fig 9A). The greatest percentage of cell clusters within the microspheres stained positive for dithizone and there was no statistical difference in the percentage of dithizone-positive clusters before and after transplantation (Fig 9B).

## Discussion

The results of the pilot biocompatibility and efficacy studies described here validate the use of PEGDA microspheres as an encapsulant to protect allogeneic cells from immune destruction, while allowing secretion of insulin from the cells to normalize blood glucose values. We have characterized fully the PEGDA formulation utilized here in previous publications showing the monodispersity, swelling ratio, diffusion characteristics and rheology [38]. In addition, we compared the results of transplants using encapsulated and unencapsulated canine islets into mice [21]. In addition, initial indications of a minimum effective dose were determined. PEG has long been utilized to improve the biocompatibility profile of other encapsulating compounds such as alginate [39, 40]. In fact, the practice continues today with the combination of PEG with other encapsulants to obtain the desired physical features such as durability while improving the biocompatibility [41, 42]. Specific to islets, PEG has been used with laborious methods that assemble individual layers of streptavidin and biotin-PEG conjugates to coat single islets [43]. The PEG coating was reported to improve glucose-stimulated insulin secretion, but the intense manufacturing procedure limits it applicability in the clinical setting.

While the use of a chemically-induced diabetic animal model is not optimal, these studies pave the way for subsequent research in spontaneously diabetic dogs, an excellent animal model for human type 1 diabetes. However, there are still major gaps in our understanding of canine diabetes and specifically characterization of canine islets. Canine islets have different features compared to human islets including a smaller size and lower density (Table 1 and Fig 1A), which is consistent with previous publications [31, 32, 44]. This may explain the nearly linear secretion of insulin in response to increasing glucose concentrations. Canine islets tend to be smaller in diameter than human or rat islets as noted in Table 1. We have previously published work correlating islet diameters to insulin secretion in various species [45, 46].

While the canine islet purity achieved was within the published ranges [27, 44, 47, 48], the islet cell viability was slightly lower compared to previously published ranges of 87% [48] to 95% [27]. However, our viability and purity values were within the acceptable range published by the Collaborative Islet Transplant Registry for humans with ranges [49]. Canine islets are notoriously difficult to isolate from the pancreas [21, 44] due partly to their small size (Tables 1 and 2) and low cell density (Fig 1A), resulting in lower initial viabilities after isolation. It is possible that factors released from dead or dying cells could attribute to some of the fibrosis noted in the surrounding tissue after transplantation, which is why the study in healthy dogs was conducted. Half of the microspheres infused into dogs #1–4 were empty while the other half contained cells. There were no differences in the amount or type of inflammatory response surrounding microspheres with and without cells (Fig 6C and 6D). Thus, lower viability values likely were not responsible for some of the foreign body response that was noted by the pathologists.

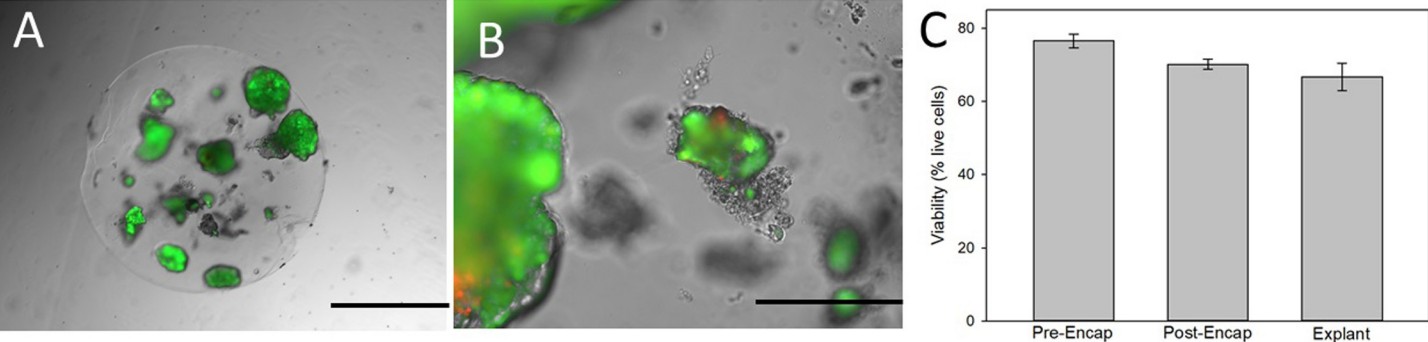

**Fig 8. Viability of cells from explanted microspheres.** A) A typical microsphere following retrieval from a diabetic dog shows intact encapsulated islets stained for viability (green). Scale bar = 500 μm. B) A higher magnification of a microspheres indicates the viability within the islets with rare red staining for dead cells, while the majority of the cells were green (alive). Scale Bar = 40 μm. C) Calcein staining was calculated as a percentage of the total cellular area and compared with the viability of the canine islets prior to encapsulation, after encapsulation (before transplant) and after the transplantation (explant). There were no statistical differences between the groups.

The study design required the administration of small doses of islets into healthy non-diabetic dogs to assess the biocompatibility of the cells plus hydrogel. The four healthy dogs received increasing doses of islets to the highest dose of 1.7 M cells/kg body weight. While the healthy dogs received lower doses of islets (Table 3), the volume of beads was significantly higher as half of the microspheres injected were empty in order to establish possible foreign body responses to the beads alone. Despite being non-diabetic, there were no hypoglycemic events measured. Importantly, the dogs had high creatinine kinase, high unconjugated bilirubin, high potassium and low calcium levels prior to the infusion of islets (Table 6). After the encapsulated islet transplant, the values either normalized or were closer to the normal values. This is noteworthy, because other microspheres encapsulants, such as polymethylmethacrylate have elicited abnormal liver enzyme values [50]. There was only a temporary increase in fibrinogen levels one week after transplants that normalized by the study termination (Table 7).

For the diabetic animals (#5–7), abnormal liver enzymes were noted prior to the transplants when the animals were receiving insulin injections to control blood glucose, but all were

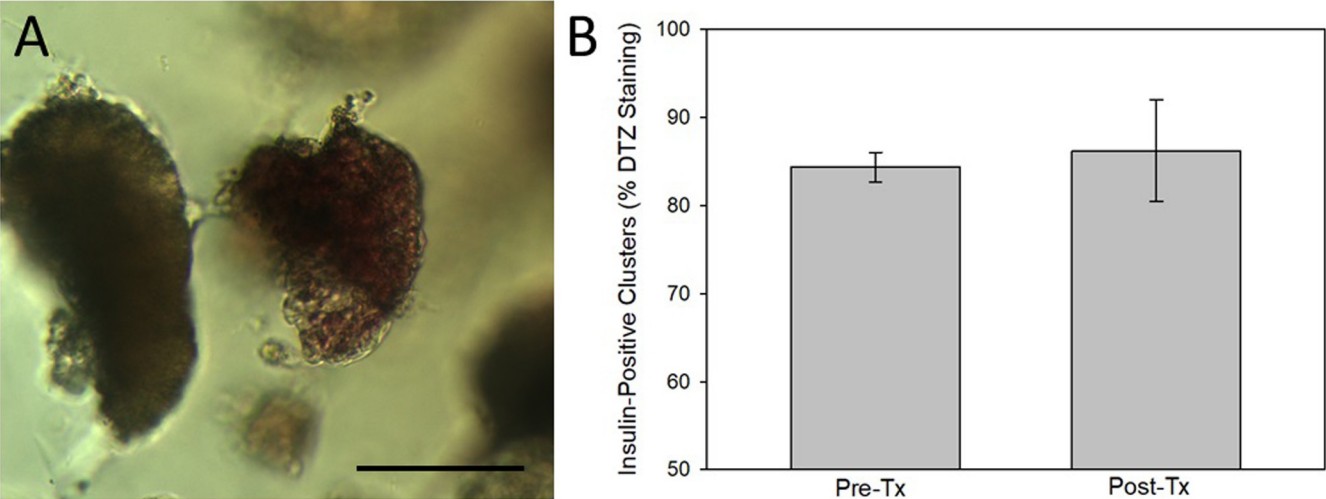

**Fig 9. Dithizone staining of explanted microspheres.** A) Example of a positively stained islet for insulin (via dithizone). Scale bar = 20 μm. B) The percentage of insulin-positive clusters (dithizone stained) were compared prior to transplantation and after. There were no differences between the groups.

normalized by the week following the transplants and remained normal through the duration of the study (Table 6). Cell counts showed low MCHCs prior to the transplants in the diabetic dogs that normalized by the end of the study (Table 7). Only fibrinogen levels were normal prior to the transplants and became or remained high after the transplants.

PEG hydrogels have been reported to elicit a foreign body response (FBR) including when using PEGDA [51–54]. In this study, retrieved microspheres at 2 or 4 weeks from healthy dogs and between 58–65 days in the diabetic dogs showed no obvious cellular overgrowth as shown in Fig 8A. At 14 days, there was some moderate multilocular inflammatory responses in the surrounding tissue showing a mild foreign body response (Fig 6A). However, 2 weeks later the local inflammation had decreased to a mild level (Fig 6B) and at 7–9 weeks in the diabetic dogs there were minimal signs of a foreign body response except for a few microspheres attached to the liver (Fig 7C). This may not be surprising, as the liver has the largest concentration of phagocytic cells in the body [55]. Further, when the microspheres clustered together rather than individual spheres within the tissue a strong foreign body response was noted (Fig 7D). The results agree with previous publications showing that a modified PEG elicited a FBR shortly after implantation that stabilized or resolved by 4 weeks [54]. In fact, all non-biological material will elicit some level of FBR, but the more severe responses will likely lead to graft failure.

## Conclusions

The PEGDA encapsulated islets provided a durable normalization of blood glucose for 60 days at the highest dose tested in agreement with our earlier mouse data showing reversal of diabetes for more than 110 days [21]. Thus, we have shown in both rodents and dogs that a long-term reversal of diabetes can be achieved with PEGDA microencapsulation. Modifications of PEGDA may further optimize its ability to enhance islet cell function while protecting the cells from the immune system, thus reducing the final dose required to achieve normoglycemia.

## Supporting information

**S1 File.**
(DOCX)

## Acknowledgments

The authors wish to thank Dr. James W. Crissman, DVM, PhD, DACVP and Dr. Laura Elcock DVM, PhD, DACVP for assistance with pathological analysis. Dr. Luke Zhang and Dr. Kevin DeDonder, DVM for assistance with animal care, Dr. Ed Robb, DVM for assistance designing the studies and for Dr. Everett Hall for assistance with canine islet cell counts.

## Author Contributions

**Conceptualization:** Stephen Harrington, Francis Karanu, Karthik Ramachandran, Lisa Stehno-Bittel.

**Data curation:** Lisa Stehno-Bittel.

**Formal analysis:** Lisa Stehno-Bittel.

**Investigation:** S. Janette Williams.

**Methodology:** Stephen Harrington, Francis Karanu, Karthik Ramachandran, S. Janette Williams.

**Project administration:** Stephen Harrington, Francis Karanu.

**Supervision:** Stephen Harrington, Francis Karanu, Lisa Stehno-Bittel.

**Writing – original draft:** Lisa Stehno-Bittel.

**Writing – review & editing:** Stephen Harrington, Francis Karanu, Karthik Ramachandran, S. Janette Williams, Lisa Stehno-Bittel.

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
