## [Decision Letter · Decision Letter 0]

14 Oct 2021

PONE-D-21-28658PEGDA microencapsulated allogeneic islets reverse canine diabetes without immunosuppression

PLOS ONE

Dear Dr. Stehno-Bittel,

Thank you for submitting your manuscript to PLOS ONE. After careful consideration, we feel that it has merit but does not fully meet PLOS ONE’s publication criteria as it currently stands. Therefore, we invite you to submit a revised version of the manuscript that addresses the points raised during the review process.

We recommend that you provide quantitative data, insulin secretion function of islets and statistical analysis concerning the overall output of the in vivo study (cell viability pre-and after transplantation, presence of insulin-positive islets after explant, ability of islets to respond to glucose stimulation after explant, average and error of glucose concentration on transplanted individuals vs controls overtime, etc). 

We look forward to receiving your revised manuscript.

Kind regards,

Seda Kizilel, PhD

Academic Editor

PLOS ONE

Journal Requirements:

2. As part of your revision, please complete and submit a copy of the Full ARRIVE 2.0 Guidelines checklist, a document that aims to improve experimental reporting and reproducibility of animal studies for purposes of post-publication data analysis and reproducibility: https://arriveguidelines.org/sites/arrive/files/Author%20Checklist%20-%20Full.pdf (PDF). Please include your completed checklist as a Supporting Information file. Note that if your paper is accepted for publication, this checklist will be published as part of your article.

 [Yes, Likarda, LLC supported the research described here. No grants funded these experiments.]. 

[I have read the journal's policy and the authors of this manuscript have the following competing interests:

SH and FK are employees of Likarda, LLC.  SJW is a former employee of Likarda, LLC. LSB and KR are employees and co-owners of Likarda, LLC. SH and KR are co-inventors on the CSS technology used to create the microspheres tested in this manuscript]. 

Reviewers' comments:

Reviewer's Responses to Questions

**Comments to the Author**

1. Is the manuscript technically sound, and do the data support the conclusions?

Reviewer #1: Partly

Reviewer #2: Partly

2. Has the statistical analysis been performed appropriately and rigorously? 

Reviewer #1: N/A

Reviewer #2: Yes

3. Have the authors made all data underlying the findings in their manuscript fully available?

Reviewer #1: Yes

Reviewer #2: Yes

4. Is the manuscript presented in an intelligible fashion and written in standard English?

Reviewer #1: Yes

Reviewer #2: Yes

5. Review Comments to the Author

Reviewer #1: The manuscript presents an interesting approach in which an oil-free method is used to produce PEGDA microgels (UV-crosslinked), and these biomaterials are used as immuneprotective carriers for the transplantation of pancreatic islets in dogs. This is a thourough study that showcases the developed materials as promising platforms for islet transplantation. The manuscript, however, lacks the presentation of data in bulk (in plots, for example), showcasing quantitative data and statistical analysis concerning the overall output of the in vivo study (cell viability pre-and after transplantation, presence of insulin-positive islets after explant, ability of islets to respond to glucose stimulation after explant, average and error of glucose concentration on transplanted individuals vs controls overtime, etc). Overall, the lack of bulk data and statistical analysis severely weakens the manuscript. Additionally, histological analysis of the tissues after transplantation is needed to exclude the formation of a fibrotic capsule (although not visible) or pro-inflammatory environment surrounding the capsules. Therefore, I believe that after data treatment and bulk statistical treament of data (and further discussion), the manuscript may be suitable for publication.

Reviewer #2: Harrington and colleagues studied microencapsulated canine islet allotransplantation and provided interesting results as pilot study. However, there are major issues needed to be addressed in this study.

1) In section of “Cytotoxicity of CSS Crosslinking Procedure”, authors performed analysis on the effect of CaCl2 concentration and presence of initiator, showed the changes on viability. For these cells, investigation of viability and functionality are crucial to see the performance of the proposed system. Because it will be a system functioning as a pancreatic endocrine tissue, changes in insulin secretion should also be reported. Viability alone will not be sufficient.

2) For different concentrations of CaCl2 and initiator, how is the size of the spheres affected?

3) How is the islet size and cell number related to the insulin secretion response? How will the stimulation index change?

4) Although explanations about histological analysis is given, figures of such study will be useful.

5) In Table 1, viabilities are around 60%-75%. Is this level acceptable for the transplant? Can authors comment on whether the possibility of factors released from that much of dead cells can contribute the immune response observed?

6) In healthy animals, as controls, microsphere only group is mentioned, but as in Table 1, healthy animals also received islet transplantation. These animals have already functional islets so there is no use of transplantation for them. But the lack of control group is problematic. If there is any data, it should be provided for a comparison.

7) Figure 3 shows post-encapsulation viability above 97.5% whereas in Table 1, it is around 60%-75%. Can authors explain the reason for such difference?

8) After which level of glucose (mg/dl) authors accepted dogs diabetic?

9) In the manuscript, authors mostly compared diabetics dogs #5-7 with healthy #1. What about #2-4? Did they show similar results with #1?

10) Changes in the blood glucose levels in healthy animals should be provided throughout the study.

11) Healthy animal group is composed female dogs and diabetic ones are male. Is there a specific reason for this?

12) In section “Explanted Microspheres”, there are no Figure 7C and Figure 7D. Figure 7C should be Figure 8B. Figure 7D should be Figure 8C.

13) When comparing groups, animals having same period of treatment should be used. For example, comparisons are made between #1 and #5-7. But post encapsulation duration is referred as nearly 1 month for #1 and for #5-7, it is about 2 months. For example, for both groups, it should be 1 month or 2 months.

14) After optimization, which condition is used to encapsulate islets for transplantation?

15) Did authors count how many islets are there in each microspheres?

16) What is the porosity of PEG hydrogel?

17) Can authors comment on reason on why #5-7 show different profiles on insulin levels (Figure 6)?

6. PLOS authors have the option to publish the peer review history of their article (what does this mean?). If published, this will include your full peer review and any attached files.

Reviewer #1: No

Reviewer #2: **Yes: **Tugba Bal

---

## [Author Response · Author response to Decision Letter 0]

21 Feb 2022

Nov 28, 2021 

Response to editors comments: PEGDA Microencapsulated Allogeneic Islets Reverse Canine Diabetes without Immunosuppression, Harrington et al.

Editor’s Comments

We recommend that you provide quantitative data, insulin secretion function of islets and statistical analysis concerning the overall output of the in vivo study (cell viability pre-and after transplantation, presence of insulin-positive islets after explant, ability of islets to respond to glucose stimulation after explant, average and error of glucose concentration on transplanted individuals vs controls overtime, etc). 

In response to the editor’s comments, we have added the following data:

A full dose-response graph of insulin secretion to glucose concentrations of unencapsulated cells has been added to the manuscript (Figure 1D). We also added the results of static insulin secretion in low and high glucose after encapsulation (Figure 2D).

In the original submission, we included figures with the twice daily non-fasting blood glucose readings for each of the diabetic dogs with the goal of providing the maximum data points for the study. In addition, we struggled with how to compare the animals on the same graph with time as the X axis, when the second transplants for dog 5 and 7 were done at different time points relative to their first transplants. However, based on the editor’s and reviewers’ comments, we appreciate that they prefer to see direct comparisons on the same graph, we have included three new graphs for Figure 4. 

First, we had discussed changes in body weight in the first submission, but did not show the data. Figure 4A now graphs the weekly body weights of the dogs. To make a valid comparison of blood glucose with the non-diabetic dogs, we included a new graph with the weekly fasting blood glucose readings (Figure 4B). Since fasting blood glucose was only taken once per week, there are no error bars on this graph. In addition, we re-graphed the original Figures 5A, B, and C into a single graph with twice weekly averages and standard error bars (new Figure 4C).

A new graph was added, comparing viability pre and post-transplantation (Figure 8C)

We were unable to obtain GSIS values from explants because the majority of microspheres were attached to tissue as shown in Figures 5A and B. In attempting to dislodge the attached spheres, they were damaged to the point that they were not valid for a GSIS study and there were insufficient numbers of free microspheres to conduct GSIS. Thus, the free microspheres were used to determine explant viability and insulin staining with dithizone. The new graph showing the percentage of insulin positive cells before and after transplantation is Figure 9B.

Additional changes were made to the manuscript in response to the reviewers’ comments including two new figures with examples of the histology results (Figures 6 & 7) . Additional data was added to the tables, including the number of clusters/spheres. 

Finally, statistical analysis was run on all comparison assays.

All changes are highlighted with track changes in the manuscript.

Taken together, these substantial changes vastly improve the completeness and the clarity of the manuscript.

Reviewers’ Comments to the Author

Reviewer #1: The manuscript presents an interesting approach in which an oil-free method is used to produce PEGDA microgels (UV-crosslinked), and these biomaterials are used as immuneprotective carriers for the transplantation of pancreatic islets in dogs. This is a thourough study that showcases the developed materials as promising platforms for islet transplantation. The manuscript, however, lacks the presentation of data in bulk (in plots, for example), showcasing quantitative data and statistical analysis concerning the overall output of the in vivo study (cell viability pre-and after transplantation, presence of insulin-positive islets after explant, ability of islets to respond to glucose stimulation after explant, average and error of glucose concentration on transplanted individuals vs controls overtime, etc). Overall, the lack of bulk data and statistical analysis severely weakens the manuscript. Additionally, histological analysis of the tissues after transplantation is needed to exclude the formation of a fibrotic capsule (although not visible) or pro-inflammatory environment surrounding the capsules. Therefore, I believe that after data treatment and bulk statistical treament of data (and further discussion), the manuscript may be suitable for publication.

We appreciate the comments made by the reviewer and agree with their perspective. However, some of the experiments that the reviewer requests are not possible in the dog model. For example, we could only retrieve the loose microspheres from the dogs after euthanasia. As shown in the original Figure 7 (now Figure 5), the vast majority of microspheres were adhered to the tissue. With the limited number of loose microspheres, we conducted viability and dithizone staining, which we included as examples in the original submission. In the revision, we have added the graphs that the reviewer requests for viability and percentage of dithizone (insulin-producing) clusters – new Figures 8C and 9B. 

Based on the reviewer’s suggestions, we created a new graph that directly compares the diabetic to non-diabetic animals’ blood glucose over time (new Figure 4B). However, it is important to point out that the study was not designed with the non-diabetic animals serving as controls for the diabetic animals. Rather the study design was two-fold. The first aim was to determine biocompatibility of the transplants in non-diabetic dogs. Second was to obtain pilot data on the efficacy of the transplants in chemically-induced diabetic dogs. In the revision, we have made a strong attempt to avoid confusion by stating more clearly the aims of the study arms and the scientific design. In the new blood glucose comparison graph (Figure 4B) it is obvious that the time monitored post-transplant is very different for the two groups because the aims were different. We also created a new graph of the twice/weekly average of non-fasting blood glucose for the diabetic animals (Figure 4C). 

Finally, statistical analysis was applied when appropriate.

Reviewer #2: Harrington and colleagues studied microencapsulated canine islet allotransplantation and provided interesting results as pilot study. However, there are major issues needed to be addressed in this study.

1) In section of “Cytotoxicity of CSS Crosslinking Procedure”, authors performed analysis on the effect of CaCl2 concentration and presence of initiator, showed the changes on viability. For these cells, investigation of viability and functionality are crucial to see the performance of the proposed system. Because it will be a system functioning as a pancreatic endocrine tissue, changes in insulin secretion should also be reported. Viability alone will not be sufficient.

We realize that we were not clear about the design and rationale of the precursor study. The original Figure 1 described the canine islets prior to encapsulation. Figure 2 summarized an optimization study using the precursors in their liquid form, not as crosslinked hydrogels. This was simply to optimize the precursor formulation. We subsequently encapsulated the cells with the optimized precursors and then tested viability, dithizone staining and insulin secretion (Figure 3). From the reviewer’s comments, we realize that the hydrogel optimization experiment did not fit with the major aims of the two studies. It was work done leading up to the canine transplants. Therefore, we have removed Figure 2 from the original submission and all associated text.

In the resubmission, we have added text to better describe the progression and rationale of the experiments found in the last paragraph of the introduction (page 6) in the methods section (page 12).

2) For different concentrations of CaCl2 and initiator, how is the size of the spheres affected?

In general, as the concentration of either CaCl2 or the initiator decrease, the size of the microspheres increases. However, we have removed the section of the manuscript that tested different calcium concentrations on cell toxicity.

3) How is the islet size and cell number related to the insulin secretion response? How will the stimulation index change?

We have published extensively on the correlations between islet size, cell number and insulin secretion and the results indicate that smaller islets release more insulin per cell. We have added a section on page 22 in the discussion, referencing our previous publications.

4) Although explanations about histological analysis is given, figures of such study will be useful.

We agree and have added histology figures for the diabetic canine experiments to the original pathologists description (new Figures 6 and 7).

5) In Table 1, viabilities are around 60%-75%. Is this level acceptable for the transplant? Can authors comment on whether the possibility of factors released from that much of dead cells can contribute the immune response observed?

The reviewer makes an excellent point and one that we have considered extensively but failed to include it in the manuscript. The most recent publication from the Collaborative Islet Transplant Registry shows that the range of islet viability in human islet transplants was from 62-98% with averages between 87 – 94%. Our canine post-encapsulation viability ranged from 67-75% (not 60-75%), well within the range of human islet transplantations. 

Canine islets are less compact and more difficult to isolate as we have shown in previous publications, resulting in viability that is on the lower end of the acceptable range. We added a new paragraph in the discussion to cover this topic and the potential impact on the immune response that was noted in our study (page 23). However, as we described in the original submission, the purpose of the biocompatibility study was to determine whether there was an impact on the surrounding tissue with empty microspheres compared to microspheres containing allogeneic islets. As shown in the new Figures 6C and D, there was no difference in the immune response surrounding empty microspheres (without cells) compared to cell-containing microspheres. This observation reduces the likelihood that dead cells caused the response, and this explanation has been added to the discussion (page 23).

6) In healthy animals, as controls, microsphere only group is mentioned, but as in Table 1, healthy animals also received islet transplantation. These animals have already functional islets so there is no use of transplantation for them. But the lack of control group is problematic. If there is any data, it should be provided for a comparison.

The reviewer is correct that the healthy dogs received empty beads and beads with islets. As stated above, that portion of the study was designed as a safety/biocompatibility study for subsequent FDA submission as preclinical data. It was never designed to serve as a control for the diabetic animals. We have attempted to make that point more clearly in the introduction and in the methods section (page 12).

The study design for the diabetic efficacy study was a pre-post design. We now show fasting blood glucose values 14-20 days prior to the transplants compared to the values up to 7 weeks after the transplants in the new Figure 4B

In addition, there is an ethical decision to be made when using chemically induced dogs as a transplant model. Knowing from historical literature that unencapsulated islets transplanted into immune-competent dogs will result in transplant failure, we chose to compare the effects of unencapsulated versus encapsulated islets in a lower animal model (mice). Those data have already been published in Harrington et al, 2020 and we have directed the readers to that publication on page 22 of this revised submission.

7) Figure 3 shows post-encapsulation viability above 97.5% whereas in Table 1, it is around 60%-75%. Can authors explain the reason for such difference?

The difference in those values is due to the experimental conditions. The calcium and the photoinitiator (Irgacure) toxicity studies were done in solution. The cells were not in fully cross-linked hydrogel microspheres, as we were testing the precursors to encapsulation. Given the confusion that the original Figure 3 caused and the fact that it did not directly fit within the aims of the studies, we have removed it and the associated text.

8) After which level of glucose (mg/dl) authors accepted dogs diabetic?

We were at fault for not providing the criteria for the diabetic state in dogs. We have corrected this error on page 13 of the current manuscript. Blood glucose in the diabetic dogs was persistent non-fasting blood glucose of over 250 mg/dL for a week.

9) In the manuscript, authors mostly compared diabetics dogs #5-7 with healthy #1. What about #2-4? Did they show similar results with #1?

We assume that the reviewer is referring to Table 7 where we only listed dogs that had abnormal CBC or hematology results. In those cases, dog #1 most commonly had out-of-range values compared to the other healthy dogs. However, in the supplemental data tables, ALL lab values (within range and out-of-range) for the 7 dogs were provided.

10) Changes in the blood glucose levels in healthy animals should be provided throughout the study.

The fasting blood glucose values for the non-diabetic dogs has been added to the results section as Figure 4B. It is important to note that this healthy animal study was designed to look at acute (2 and 4 week) biocompatibility rather than as a comparison for the diabetic treated animals. Thus, values for those animal only go through weeks 2-4 post-transplant.

11) Healthy animal group is composed female dogs and diabetic ones are male. Is there a specific reason for this?

The sex differences in the study were simply due to the animals that were available at the CROs where the studies were conducted. We are planning to repeat this study using differentiated induced pluripotent stem cells rather than canine islets. In that subsequent study, we will be utilizing equal numbers of male and female animals.

12) In section “Explanted Microspheres”, there are no Figure 7C and Figure 7D. Figure 7C should be Figure 8B. Figure 7D should be Figure 8C.

Thank you for the correction. We have made these edits.

13) When comparing groups, animals having same period of treatment should be used. For example, comparisons are made between #1 and #5-7. But post encapsulation duration is referred as nearly 1 month for #1 and for #5-7, it is about 2 months. For example, for both groups, it should be 1 month or 2 months.

Again, the confusion is our fault for not clearly describing the aims and experimental design of the two separate study arms. We have added text in both the abstract, introduction, methods and discussion to clarify this point.

14) After optimization, which condition is used to encapsulate islets for transplantation?

As stated in the methods section, 100 mM calcium with PEGDA and 0.025% irgacure were utilized for the formulation of the microspheres. However, we have removed the optimization data from the revision.

15) Did authors count how many islets are there in each microspheres?

Based on the reviewer’s comments, we analyzed the saved images of the microspheres prior to transplantation and counted the islets/microsphere. These data have been added to the manuscript in Tables 1 and 2 with along with the statistical analysis.

16) What is the porosity of PEG hydrogel?

We have previously published the characteristics of the PEGDA microspheres in Harrington et al, 2020 and Hamilton et al, 2021. There we directly measured the functional porosity by quantifying the diffusion rate of molecules of various sizes out of the impregnated microspheres. Statements directing the reader to the previous characterization was added to page 22 of the discussion.

17) Can authors comment on reason on why #5-7 show different profiles on insulin levels (Figure 6)?

The original Figure 6 did not include any data related to insulin levels. The Figures only showed the daily blood glucose values. The author may have confused the text with each graph which stated the dose (cells/kg) for each transplant to remind the reader that the 3 dogs did not receive the same number of islets. 

Now that the blood glucose data has been combined into 2 figures, we do not list the cells/kg on the figure, although it is included in Table 3. In the first submission, we included the insulin requirement pre- and post-transplant in Table 4. For the resubmission, we also added the insulin administration for the first 4 dogs, which was of course zero.

---

## [Decision Letter · Decision Letter 1]

18 Apr 2022

PEGDA microencapsulated allogeneic islets reverse canine diabetes without immunosuppression

PONE-D-21-28658R1

Dear Dr. Lisa Stehno-Bittel,

We’re pleased to inform you that your manuscript has been judged scientifically suitable for publication and will be formally accepted for publication once it meets all outstanding technical requirements.

Kind regards,

Seda Kizilel, PhD

Academic Editor

PLOS ONE

Additional Editor Comments (optional):

Reviewers' comments:

Reviewer's Responses to Questions

**Comments to the Author**

1. If the authors have adequately addressed your comments raised in a previous round of review and you feel that this manuscript is now acceptable for publication, you may indicate that here to bypass the “Comments to the Author” section, enter your conflict of interest statement in the “Confidential to Editor” section, and submit your "Accept" recommendation.

Reviewer #1: All comments have been addressed

Reviewer #2: All comments have been addressed

2. Is the manuscript technically sound, and do the data support the conclusions?

Reviewer #1: Yes

Reviewer #2: Yes

3. Has the statistical analysis been performed appropriately and rigorously? 

Reviewer #1: Yes

Reviewer #2: Yes

4. Have the authors made all data underlying the findings in their manuscript fully available?

Reviewer #1: (No Response)

Reviewer #2: Yes

5. Is the manuscript presented in an intelligible fashion and written in standard English?

Reviewer #1: Yes

Reviewer #2: Yes

6. Review Comments to the Author

Reviewer #1: (No Response)

Reviewer #2: Authors addressed all my comments and I would to thank for their effort. All a final comment, on tables, plus sign should be plus minus sign.

7. PLOS authors have the option to publish the peer review history of their article (what does this mean?). If published, this will include your full peer review and any attached files.

Reviewer #1: No

Reviewer #2: No

---

## [Editor Report · Acceptance letter]

16 May 2022

PONE-D-21-28658R1 

PEGDA Microencapsulated Allogeneic Islets Reverse Canine Diabetes without Immunosuppression 

Dear Dr. Stehno-Bittel:

I'm pleased to inform you that your manuscript has been deemed suitable for publication in PLOS ONE. Congratulations! Your manuscript is now with our production department. 

Kind regards, 

on behalf of

Dr. Seda Kizilel 

Academic Editor

PLOS ONE